# The Forget-me-not Process

**Kieran Milan**[†]**, Joel Veness**[†]**, James Kirkpatrick, Demis Hassabis**
Google DeepMind
{kmilan,aixi,kirkpatrick,demishassabis}@google.com

**Anna Koop, Michael Bowling**
University of Alberta
{anna,bowling}@cs.ualberta.ca

## Abstract

We introduce the Forget-me-not Process, an efficient, non-parametric meta-algorithm for online probabilistic sequence prediction for piecewise stationary, repeating sources. Our method works by taking a Bayesian approach to partitioning a stream of data into postulated task-specific segments, while simultaneously building a model for each task. We provide regret guarantees with respect to piecewise stationary data sources under the logarithmic loss, and validate the method empirically across a range of sequence prediction and task identification problems.

## 1 Introduction

Modeling non-stationary temporal data sources is a fundamental problem in signal processing, statistical data compression, quantitative finance and model-based reinforcement learning. One widely-adopted and successful approach has been to design meta-algorithms that automatically generalize existing stationary learning algorithms to various non-stationary settings. In this paper we introduce the Forget-me-not Process, a probabilistic meta-algorithm that provides the ability to model the class of memory bounded, piecewise-repeating sources given an arbitrary, probabilistic memory bounded stationary model.

The most well studied class of probabilistic meta-algorithms are those for piecewise stationary sources, which model data sequences with abruptly changing statistics. Almost all meta-algorithms for abruptly changing sources work by performing Bayesian model averaging over a class of hypothesized temporal partitions. To the best of our knowledge, the earliest demonstration of this fundamental technique was [21], for the purpose of data compression; closely related techniques have gained popularity within the machine learning community for change point detection [1] and have been proposed by neuroscientists as a mechanism by which humans deal with open-ended environments composed of multiple distinct tasks [4–6]. One of the reasons for the popularity of this approach is that the temporal structure can be exploited to make exact Bayesian inference tractable via dynamic programming; in particular inference over all possible temporal partitions of $n$ data points results in an algorithm of $O(n^2)$ time complexity and $O(n)$ space complexity [21, 1]. Many variants have been proposed in the literature [20, 11, 10, 17], which trade off predictive accuracy for improved time and space complexity; in particular the Partition Tree Weighting meta-algorithm [17] has $O(n \log n)$ time and $O(\log n)$ space complexity, and has been shown empirically to exhibit superior performance versus other low-complexity alternatives on piecewise stationary sources.

A key limitation of these aforementioned techniques is that they can perform poorly when there exist multiple segments of data that are similarly distributed. For example, consider data generated according to the schedule depicted in Figure 1. For all these methods, once a change-point occurs, the base (stationary) model is invoked from scratch, even if the task repeats, which is clearly undesirable

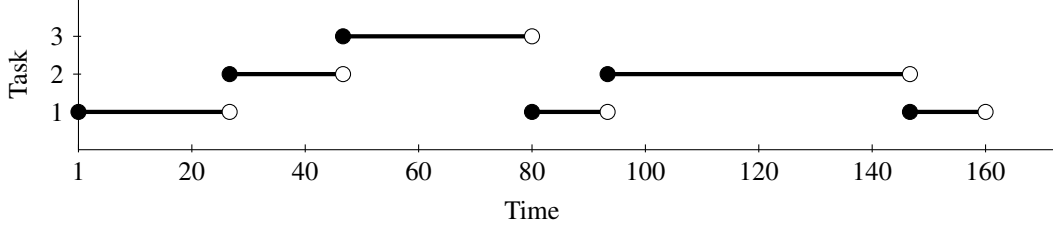

Figure 1: An example task segmentation.

in many situations of interest. Our main contribution in this paper is to introduce the Forget-me-not Process, which has the ability to avoid having to relearn repeated tasks, while still maintaining essentially the same theoretical performance guarantees as Partition Tree Weighting on piecewise stationary sources.

## 2 Preliminaries

We now introduce some notation and necessary background material.

**Sequential Probabilistic Data Generators.** We begin with some terminology for sequential, probabilistic data generating sources. An alphabet is a finite non-empty set of symbols, which we will denote by $\mathcal{X}$. A string $x_1 x_2 \ldots x_n \in \mathcal{X}^n$ of length $n$ is denoted by $x_{1:n}$. The prefix $x_{1:j}$ of $x_{1:n}$, where $j \leq n$, is denoted by $x_{\leq j}$ or $x_{<j+1}$. The empty string is denoted by $\epsilon$ and we define $\mathcal{X}^* = \{\epsilon\} \cup \bigcup_{i=1}^{\infty} \mathcal{X}^i$. Our notation also generalizes to out of bounds indices; that is, given a string $x_{1:n}$ and an integer $m > n$, we define $x_{1:m} := x_{1:n}$ and $x_{m:n} := \epsilon$. The concatenation of two strings $s, r \in \mathcal{X}^*$ is denoted by $sr$. Unless otherwise specified, base 2 is assumed for all logarithms.

A sequential probabilistic data generating source $\rho$ is defined by a sequence of probability mass functions $\rho_n : \mathcal{X}^n \to [0, 1]$, for all $n \in \mathbb{N}$, satisfying the constraint that $\rho_n(x_{1:n}) = \sum_{y \in \mathcal{X}} \rho_{n+1}(x_{1:n}y)$ for all $x_{1:n} \in \mathcal{X}^n$, with base case $\rho_0(\epsilon) = 1$. From here onwards, whenever the meaning is clear from the argument to $\rho$, the subscripts on $\rho$ will be dropped. Under this definition, the conditional probability of a symbol $x_n$ given previous data $x_{<n}$ is defined as $\rho(x_n \mid x_{<n}) := \rho(x_{1:n})/\rho(x_{<n})$ provided $\rho(x_{<n}) > 0$, with the familiar chain rule $\rho(x_{i:j} \mid x_{<i}) = \prod_{k=i}^{j} \rho(x_k \mid x_{<k})$ applying as usual. Notice too that a new sequential probabilistic data generating source $\nu$ can be obtained from an existing source $\rho$ by conditioning on a fixed sequence of input data. More explicitly, given a string $s \in \mathcal{X}^*$, one can define $\nu(x_{1:n}) := \rho(x_{1:n} \mid s)$ for all $n$; we will use the notation $\rho[s]$ to compactly denote such a derived probabilistic data generating source.

**Temporal Partitions, Piecewise Sources and Piecewise-repeating sources.** We now introduce some notation to formally describe temporal partitions and piecewise sources. A segment is a tuple $(a, b) \in \mathbb{N} \times \mathbb{N}$ with $a \leq b$. A segment $(a, b)$ is said to overlap with another segment $(c, d)$ if there exists an $i \in \mathbb{N}$ such that $a \leq i \leq b$ and $c \leq i \leq d$. A temporal partition $\mathcal{P}$ of a set of time indices $S = \{1, 2, \ldots n\}$, for some $n \in \mathbb{N}$, is a set of non-overlapping segments such that for all $x \in \mathcal{S}$, there exists a segment $(a, b) \in \mathcal{P}$ such that $a \leq x \leq b$. We also use the overloaded notation $\mathcal{P}(a, b) := \{(c, d) \in \mathcal{P} : a \leq c \leq d \leq b\}$ to denote the set of segments falling inclusively within the range $(a, b)$. Finally, $\mathcal{T}_n$ will be used to denote the set of all possible temporal partitions of $\{1, 2, \ldots, n\}$.

We can now define a piecewise data generating source $\mu_{\mathcal{P}}^h$ in terms of a partition $\mathcal{P} = \{(a_1, b_1), (a_2, b_2), \ldots\}$ and a set of probabilistic data generating sources $\{\mu^1, \mu^2, \ldots\}$, such that for all $n \in \mathbb{N}$, for all $x_{1:n} \in \mathcal{X}^n$,

$$\mu_{\mathcal{P}}^h(x_{1:n}) := \prod_{(a,b) \in \mathcal{P}_n} \mu^{h(a)}(x_{a:b}),$$

where $\mathcal{P}_n := \{(a, b) \in \mathcal{P} : a \leq n\}$ and $h : \mathbb{N} \to \mathbb{N}$ is a task assignment function that maps segment beginnings to task identifiers.

A piecewise *repeating* data generating source is a special case of a piecewise data generating source that satisfies the additional constraint that $\exists a, c \in \{x : (x, y) \in \mathcal{P}\}$ such that $a \neq c$ and $h(a) = h(c)$.

In terms of modeling a piecewise repeating source, there are three key unknowns: the partition which defines the location of the change points, the task assignment function, and the model for each individual task.

**Bayesian Sequence Prediction.** A fundamental technique for constructing algorithms that work well under the logarithmic loss is Bayesian model averaging. We now provide a short overview sufficient for the purposes of this paper; for more detail, we recommend the work of [12] and [14].

Given a non-empty discrete set of probabilistic data generating sources $\mathcal{M} := \{\rho_1, \rho_2, \dots\}$ and a prior weight $w_0^\rho > 0$ for each $\rho \in \mathcal{M}$ such that $\sum_{\rho \in \mathcal{M}} w_0^\rho = 1$, the Bayesian mixture predictor is defined in terms of its marginal by $\xi(x_{1:n}) := \sum_{\rho \in \mathcal{M}} w_0^\rho \rho(x_{1:n})$. The predictive probability is thus given by the ratio of the marginals $\xi(x_n \mid x_{<n}) = \xi(x_{1:n}) / \xi(x_{<n})$. The predictive probability can also be expressed in terms of a convex combination of conditional model predictions, with each model weighted by its posterior probability. More explicitly,

$$\xi(x_n \mid x_{<n}) = \frac{\sum\limits_{\rho \in \mathcal{M}} w_0^\rho \, \rho(x_{1:n})}{\sum\limits_{\rho \in \mathcal{M}} w_0^\rho \, \rho(x_{<n})} = \sum_{\rho \in \mathcal{M}} w_{n-1}^\rho \, \rho(x_n \mid x_{<n}), \text{ where } w_{n-1}^\rho := \frac{w_0^\rho \, \rho(x_{<n})}{\sum\limits_{\nu \in \mathcal{M}} w_0^\nu \, \nu(x_{<n})}.$$

A fundamental property of Bayesian mixtures is that if there exists a model $\rho^* \in \mathcal{M}$ that predicts well, then $\xi$ will predict well since the cumulative loss satisfies

$$-\log \xi(x_{1:n}) = -\log \sum_{\rho \in \mathcal{M}} w_0^\rho \, \rho(x_{1:n}) \leq -\log w_0^{\rho^*} - \log \rho^*(x_{1:n}). \tag{1}$$

Equation 1 implies that a constant regret is suffered when using $\xi$ in place of the best (in hindsight) model within $\mathcal{M}$.

## 3 The Forget-me-not Process

We now introduce the Forget-me-not Process (FMN), a meta-algorithm designed to better model piecewise-repeating data generating sources. As FMN is a meta-algorithm, it takes as input a base model, which we will hereby denote as $\nu$. At a high level, the main idea is to extend the Partition Tree Weighting [17] algorithm to incorporate a memory of previous model states, which is used to improve performance on repeated tasks. More concretely, our construction involves defining a two-level hierarchical process, with each level performing exact Bayesian model averaging. The first level will perform model averaging over a set of postulated segmentations of time, using the Partition Tree Weighting technique. The second level will perform model averaging over a *growing* set of stored base model states. We describe each level in turn before describing how to combine these ideas into the Forget-me-not Process.

**Averaging over Temporal Segmentations.** We now define the class of *binary temporal partitions*, which will correspond to the set of temporal partitions we perform model averaging over in the first level of our hierarchical model. Although more restrictive than the class of all possible temporal partitions, binary temporal partitions possess important computational advantages.

**Definition 1.** *Given a depth parameter $d \in \mathbb{N}$ and a time $t \in \mathbb{N}$, the set $\mathcal{C}_d(t)$ of all binary temporal partitions from $t$ is recursively defined by*

$$\mathcal{C}_d(t) := \left\{ \{(t, t + 2^d - 1)\} \right\} \cup \left\{ \mathcal{S}_1 \cup \mathcal{S}_2 : \mathcal{S}_1 \in \mathcal{C}_{d-1}(t), \mathcal{S}_2 \in \mathcal{C}_{d-1}\left(t + 2^{d-1}\right) \right\},$$

*with $\mathcal{C}_0(t) := \left\{ \{(t, t)\} \right\}$. We also define $\mathcal{C}_d := \mathcal{C}_d(1)$.*

Each binary temporal partition can be naturally mapped onto a tree structure known as a partition tree; for example, Figure 2 shows the collection of partition trees represented by $\mathcal{C}_2$; the leaves of each tree correspond to the segments within each particular partition. There are two important properties of binary temporal partition trees. The first is that there always exists a partition $\mathcal{P}' \in \mathcal{C}_d$ which is close to any temporal partition $\mathcal{P}$, in the sense that $\mathcal{P}'$ always starts a new segment whenever $\mathcal{P}$ does, and $|\mathcal{P}'| \leq |\mathcal{P}|(\lceil \log n \rceil + 1)$ [17, Lemma 2]. The second is that exact Bayesian model averaging can be performed efficiently with an appropriate choice of prior. This is somewhat surprising, since the

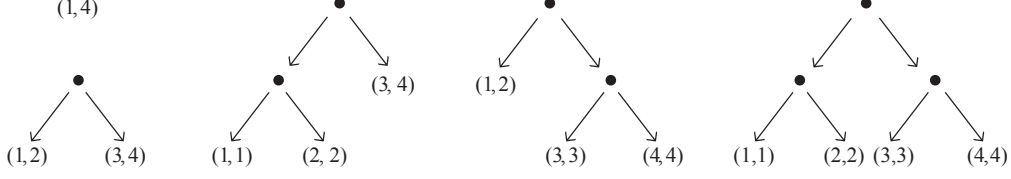

Figure 2: The set $\mathcal{C}_2$ represented as a collection of temporal partition trees.

number of binary temporal partitions $|\mathcal{C}_d|$ grows double exponentially in $d$. The trick is to define, given a data sequence $x_{1:n}$, the Bayesian mixture

$$\text{PTW}_d(x_{1:n}) := \sum_{\mathcal{P} \in \mathcal{C}_d} 2^{-\Gamma_d(\mathcal{P})} \prod_{(a,b) \in \mathcal{P}} \rho(x_{a:b}), \tag{2}$$

where $\Gamma_d(\mathcal{P})$ gives the number of nodes in the partition tree associated with $\mathcal{P}$ that have a depth less than $d$ and $\rho$ denotes the base model to the PTW process. This prior weighting is identical to how the Context Tree Weighting method [22] weighs over tree structures, and is an application of the general technique used by the class of Tree Experts described in Section 5.3 of [3]. It is a valid prior, as one can show $\sum_{\mathcal{P} \in \mathcal{C}_d} 2^{-\Gamma_d(\mathcal{P})} = 1$ for all $d \in \mathbb{N}$. A direct computation of Equation 2 is clearly intractable, but we can make use of the tree structured prior to recursively decompose Equation 2 using the following lemma.

**Lemma 1** (*Veness et al. [17]*). *For any depth $d \in \mathbb{N}$, for all $x_{1:n} \in \mathcal{X}^n$ satisfying $n \leq 2^d$,*

$$\text{PTW}_d(x_{1:n}) = \tfrac{1}{2}\rho(x_{1:n}) + \tfrac{1}{2}\text{PTW}_{d-1}(x_{1:k})\,\text{PTW}_{d-1}(x_{k+1:n}),$$

*where $k = 2^{d-1}$.*

**Averaging over Previous Model States given a Known Temporal Partition.** Given a data sequence $x_{1:n} \in \mathcal{X}^n$, a base model $\rho$ and a temporal partition $\mathcal{P} := \{(a_1, b_1), \ldots, (a_m, b_m)\}$ satisfying $\mathcal{P} \in \mathcal{T}_n$, consider a sequential probabilistic model defined by

$$\pi_{\mathcal{P}}(x_{1:n}) := \prod_{i=1}^{|\mathcal{P}|} \left( \sum_{\rho \in \mathcal{M}_i} \tfrac{1}{|\mathcal{M}_i|}\, \rho(x_{a_i:b_i}) \right),$$

where $\mathcal{M}_1 := \{\rho\}$ and $\mathcal{M}_i := \mathcal{M}_{i-1} \cup \{\rho\,[x_{a_i:b_i}]\}_{\rho \in \mathcal{M}_{i-1}}$ for $1 < i \leq |\mathcal{P}|$.

Here, whenever the $i$th segment of data is seen, each model in $\mathcal{M}_i$ is given the option of either ignoring or adapting to this segment's data, which implies $|\mathcal{M}_i| = 2\,|\mathcal{M}_{i-1}|$. Using an argument similar to Equation 1, and letting $x_{<t}^{h(t)}$ denote the subsequence of $x_{<t}$ generated by $\mu^{h(t)}$, we can see that the cumulative loss when the data is generated by a piecewise-repeating source $\mu_{\mathcal{P}}^h$ is bounded by

$$-\log \pi_{\mathcal{P}}(x_{1:n}) = -\log \prod_{i=1}^{|\mathcal{P}|} \left( \sum_{\rho \in \mathcal{M}_i} \tfrac{1}{|\mathcal{M}_i|}\, \rho(x_{a_i:b_i}) \right) = -\log \prod_{i=1}^{|\mathcal{P}|} \left( \sum_{\rho \in \mathcal{M}_i} 2^{-i+1}\, \rho(x_{a_i:b_i}) \right)$$

$$\leq -\log \prod_{i=1}^{|\mathcal{P}|} 2^{-i+1}\, \rho\!\left( x_{a_i:b_i} \mid x_{<a_i}^{h(a_i)} \right) = \frac{|\mathcal{P}|^2 - |\mathcal{P}|}{2} - \log \prod_{i=1}^{|\mathcal{P}|} \rho\!\left( x_{a_i:b_i} \mid x_{<a_i}^{h(a_i)} \right). \tag{3}$$

Roughly speaking, this bound implies that $\pi_{\mathcal{P}}(x_{1:n})$ will perform almost as well as if we knew $h(\cdot)$ in advance, provided the number of segments grows $o(\sqrt{n})$. The two main drawbacks with this approach are that: a) computing $\pi_{\mathcal{P}}(x_{1:n})$ takes time exponential in $|\mathcal{P}|$; and b) a regret of $(|\mathcal{P}|^2 - |\mathcal{P}|)/2$ seems overly large in cases where the source isn't repeating. These problems can be rectified with the following modified process,

$$\nu_{\mathcal{P}}(x_{1:n}) := \prod_{i=1}^{|\mathcal{P}|} \left( \frac{1}{2}\rho(x_{a_i:b_i}) + \frac{1}{2} \sum_{\rho' \in \mathcal{M}_i \setminus \{\rho\}} \tfrac{1}{|\mathcal{M}_i|-1}\, \rho'(x_{a_i:b_i}) \right) \tag{4}$$

where now $\mathcal{M}_1 := \{\rho\}$ and $\mathcal{M}_i := \mathcal{M}_{i-1} \cup \left\{ \rho^*[x_{a_i:b_i}] \,\middle|\, \rho^* = \text{argmax}_{\rho \in \mathcal{M}_{i-1}} \{\rho(x_{a_i:b_i})\} \right\}$.

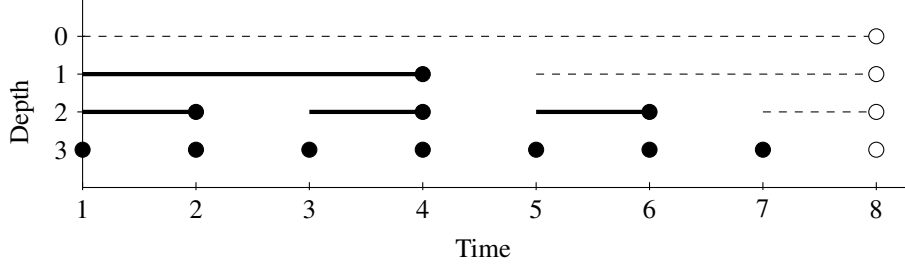

Figure 3: A graphical depiction of the Forget Me Not process ($d = 3$) after processing 7 symbols.

With this modified definition of $\mathcal{M}_i$, where the argmax implements a greedy approximation (ties are broken arbitrarily), $|\mathcal{M}_i|$ now grows linearly with the number of segments, and thus the overall time to compute $\nu_{\mathcal{P}}(x_{1:n})$ is $O(|\mathcal{P}|\,n)$ assuming the base model runs in linear time. Although heuristic, this approximation is justified provided that $\rho[\epsilon]$ assigns the highest probability out of any model in $\mathcal{M}_i$ whenever a task is seen for the first time, and that a model trained on $k$ segments for a given task is always better than a model trained on less than $k$ segments for the same task (or a model trained on any number of other tasks). Furthermore, using a similar dominance argument to Equations 1 and 3, the cost of not knowing $h(\cdot)$ with respect to piecewise non-repeating sources is now $|\mathcal{P}|$ vs $O(|\mathcal{P}|^2)$.

**Averaging over Binary Temporal Segmentations and Previous Model States.** This section describes how to hierarchically combine the PTW and $\nu_{\mathcal{P}}$ models to give rise to the Forget Me Not process. Our goal will be to perform model averaging over both binary temporal segmentations and previous model states. This can be achieved by instantiating the PTW meta-algorithm with a *sequence* of time dependent base models similar in spirit to $\nu_{\mathcal{P}}$.

Intuitively, this requires modifying the definition of $\mathcal{M}_i$ so that the best performing model state, for any completed segment within the PTW process, is available for future predictions. For example, Figure 3 provides a graphical depiction of our desired $\text{FMN}_3$ process after processing 7 symbols. The dashed segments ending in unfilled circles describe the segments whose set of base models are contributing to the predictive distribution at time 8. The solid-line segments denote previously completed segments for which we want the best performing model state to be remembered and made available to segments starting at later times. A solid circle indicates a time where a model is added to the pool of available models; note that now multiple models can be added at any particular time.

We now formalize the above intuitions. Let $\mathcal{B}_t := \{(a, b) \in \mathcal{C}_d : b = t\}$ be the set of segments ending at time $t \le 2^d$. Given an an arbitrary string $s \in \mathcal{X}^*$, our desired sequence of base models is given by

$$\nu_t(s) := \frac{1}{2}\rho(s) + \frac{1}{2} \sum_{\rho' \in \mathcal{M}_t \setminus \{\rho\}} \frac{1}{|\mathcal{M}_t| - 1}\, \rho'(s), \tag{5}$$

with the model pool defined by $\mathcal{M}_1 := \{\rho\}$ and

$$\mathcal{M}_t := \mathcal{M}_{t-1} \cup \bigcup_{(a,b) \in \mathcal{B}_{t-1}} \left\{ \rho^*[s_{a:b}] \,\middle|\, \rho^* = \operatorname*{argmax}_{\rho \in \mathcal{M}_a} \{\rho(s_{a:b})\} \right\} \quad \text{for } t > 1. \tag{6}$$

Finally, substituting Equation 5 in for the base model of PTW yields our Forget Me Not process

$$\text{FMN}_d(x_{1:n}) := \sum_{\mathcal{P} \in \mathcal{C}_d} 2^{-\Gamma_d(\mathcal{P})} \prod_{(a,b) \in \mathcal{P}_n} \nu_a(x_{a:b}). \tag{7}$$

**Algorithm.** Algorithm 1 describes how to compute the marginal probability $\text{FMN}_d(x_{1:n})$. The $r_j$ variables store the segment start times for the unclosed segments at depth $j$; the $b_j$ variables implement a dynamic programming caching mechanism to speed up the PTW computation as explained in Section 3.3 of [17]; the $w_j$ variables hold intermediate results needed to apply Lemma 1. The Most Significant Changed Bit routine $\text{MSCB}_d(t)$, invoked at line 4, is used to determine the range of segments ending at the current time $t$, and is defined for $t > 1$ as the number of bits to the left of the most significant location at which the $d$-bit binary representations of $t-1$ and $t-2$ differ, with $\text{MSCB}_d(1) := 0$ for all $d \in \mathbb{N}$. For example, in Figure 3, at $t = 5$, before processing $x_5$, we need to deal with the segments

---

**Algorithm 1** FORGET-ME-NOT - $\text{FMN}_d(x_{1:n})$

---

**Require:** A depth parameter $d \in \mathbb{N}$, and a base probabilistic model $\rho$
**Require:** A data sequence $x_{1:n} \in \mathcal{X}^n$ satisfying $n \leq 2^d$

1: $b_j \leftarrow 1, w_j \leftarrow 1, r_j \leftarrow 1$, for $0 \leq j \leq d$
2: $\mathcal{M} \leftarrow \{\rho\}$

3: **for** $t = 1$ to $n$ **do**

4:      $i \leftarrow \text{MSCB}_d(t)$
5:      $b_i \leftarrow w_{i+1}$

6:      **for** $j = i + 1$ to $d$ **do**
7:           $\mathcal{M} \leftarrow \text{UPDATEMODELPOOL}(\nu_{r_j}, x_{r_j:t-1})$
8:           $w_j \leftarrow 1, b_j \leftarrow 1, r_j \leftarrow t$
9:      **end for**

10:     $w_d \leftarrow \nu_{r_d}(x_{r_d:t})$
11:     **for** $i = d - 1$ to $0$ **do**
12:          $w_i \leftarrow \frac{1}{2}\nu_{r_i}(x_{r_i:t}) + \frac{1}{2}w_{i+1}b_i$
13:     **end for**

14: **end for**

15: **return** $w_0$

---

$(1,4), (3,4), (4,4)$ finishing. The method UPDATEMODELPOOL applies Equation 6 to remember the best performing model in the mixture $\nu_{r_j}$ on the completed segment $(r_j, t - 1)$. Lines 11 to 13 invoke Lemma 1 from bottom-up, to compute the desired marginal probability $\text{FMN}_d(x_{1:n}) = w_0$.

(*Space and Time Overhead*) Under the assumption that each base model conditional probability can be obtained in $O(1)$ time, the time complexity to process a sequence of length $n$ is $O(nk \log n)$, where $k$ is an upper bound on $|\mathcal{M}|$. The $n \log n$ factor is due to the number of iterations in the inner loops on Lines 6 to 9 and Lines 11 to 13 being upper bounded by $d + 1$. The $k$ factor is due to the cost of maintaining the $v_t$ terms for the segments which have not yet closed. An upper bound on $k$ can be obtained from inspection of Figure 3, where if we set $n = 2^d$, we have that the number of completed segments is given by $\sum_{i=0}^{d} 2^i = 2^{d+1} - 1 = 2n + 1 = O(n)$; thus the time complexity is $O(n^2 \log n)$. The space overhead is $O(k \log n)$, due to the $O(\log n)$ instances of Equation 5.

(*Complexity Reducing Operations*) For many applications of interest, a running time of $O(n^2 \log n)$ is unacceptable. A workaround is to fix $k$ in advance and use a model replacement strategy that enforces $|\mathcal{M}| \leq k$ via a modified UPDATEMODELPOOL routine; this reduces the time complexity to $O(nk \log n)$. We found the following heuristic scheme to be effective in practice: when a segment $(a, b)$ closes, the best performing model $\rho^* \in \mathcal{M}_a$ for this segment is identified. Now, 1) letting $y_\rho*$ denote a uniform sub-sample of the data used to train $\rho^*$, if $\log \rho^*[x_{a:b}](y_\rho*) - \log \rho^*(y_\rho*) > \alpha$ then *replace* $\rho^*$ with $\rho^*[x_{a:b}]$ in $\mathcal{M}$; else 2) if a uniform Bayes mixture $\xi$ over $\mathcal{M}$ assigns sufficiently higher probability to a uniform sub-sample $s$ of $x_{a:b}$ than $\rho^*$ does, that is $\log \xi(s) - \log \rho^*(s) > \beta$, then leave $\mathcal{M}$ unchanged; else 3) add $\rho^*[x_{a:b}]$ to $\mathcal{M}$; if $|\mathcal{M}| > k$, remove the oldest model in $\mathcal{M}$. This requires choosing hyperparameters $\alpha, \beta \in \mathbb{R}$ and appropriate constant sub-sample sizes. Step 1 avoids adding multiple models for the same task; Step 2 avoids adding a redundant model to the model pool. Note that the per model and per segment sub-samples can be efficiently maintained online using reservoir sampling [19]. As a further complexity reducing operation, one can skip calls to UPDATEMODELPOOL unless $(b - a + 1) \geq 2^c$ for some $c < d$.

(*Strongly Online Prediction*) A strongly online FMN process, where one does not need to fix a $d$ in advance such that $n \leq 2^d$, can be obtained by defining $\text{FMN}(x_{1:n}) := \prod_{i=1}^{n} \text{FMN}_{\lceil \log i \rceil}(x_i \mid x_{<i})$, and efficiently computed in the same manner as for PTW, with a similar loss bound $-\log \text{FMN}(x_{1:n}) \leq -\log \text{FMN}_d(x_{1:n}) + \lceil \log n \rceil (\log 3 - 1)$ following trivially from Theorem 2 in [17].

**Theoretical properties.** We now show that the Forget Me Not process is competitive with any piecewise stationary source, provided the base model enjoys sufficiently strong regret guarantees on

non-piecewise sources. Note that provided $c = 0$, Proposition 1 also holds when the complexity reducing operations are used. While the following regret bound is of the same asymptotic order as PTW for piecewise stationary sources, note that it is no tighter for sources that repeat; we will later explore the advantage of the FMN process on repeating sources experimentally.

**Proposition 1.** *For all $n \in \mathbb{N}$, using* FMN *with $d = \lceil \log n \rceil$ and a base model $\rho$ whose redundancy is upper bounded by a non-negative, monotonically non-decreasing, concave function $g : \mathbb{N} \to \mathbb{R}$ with $g(0) = 0$ on some class $\mathcal{G}$ of bounded memory data generating sources, the regret*

$$\log \left( \frac{\mu_{\mathcal{P}}^h(x_{1:n})}{\text{FMN}_d(x_{1:n})} \right) \leq 2|\mathcal{P}_n| \left( \lceil \log n \rceil + 1 \right) + |\mathcal{P}_n| \, g \left( \left\lceil \frac{n}{|\mathcal{P}_n|(\lceil \log n \rceil + 1)} \right\rceil \right) \left( \lceil \log n \rceil + 1 \right) + |\mathcal{P}_n|,$$

*where $\mu$ is a piecewise stationary data generating source, and the data in each of the stationary regions $\mathcal{P} \in \mathcal{T}_n$ is distributed according to some source in $\mathcal{G}$.*

*Proof.* First observe that for all $x_{1:n} \in \mathcal{X}^n$ we can lower bound the probability

$$\text{FMN}_d(x_{1:n}) = \sum_{\mathcal{P} \in \mathcal{C}_d} 2^{-\Gamma_d(\mathcal{P})} \prod_{(a,b) \in \mathcal{P}_n} \nu_a(x_{a:b}) \geq \sum_{\mathcal{P} \in \mathcal{C}_d} 2^{-\Gamma_d(\mathcal{P})} \prod_{(a,b) \in \mathcal{P}_n} \tfrac{1}{2} \rho(x_{a:b})$$

$$= 2^{-|\mathcal{P}_n|} \sum_{\mathcal{P} \in \mathcal{C}_d} 2^{-\Gamma_d(\mathcal{P})} \prod_{(a,b) \in \mathcal{P}_n} \rho(x_{a:b}) = 2^{-|\mathcal{P}_n|} \, \text{PTW}_d(x_{1:n}).$$

Hence we have that $-\log \text{FMN}_d(x_{1:n}) \leq |\mathcal{P}| - \log \text{PTW}_d(x_{1:n})$. The proof is completed by using Theorem 1 from [17] to upper bound $-\log \text{PTW}_d(x_{1:n})$. $\square$

## 4  Experimental Results

We now report some experimental results with the FMN algorithm across three test domains. The first two domains, *The Mysterious Bag of Coins* and *A Fistful of Digits*, are repeating sequence prediction tasks. The final domain, *Continual Atari 2600 Task Identification*, is a video stream of game-play from a collection of Atari games provided by the ALE [2] framework; here we qualitatively assess the capabilities of the FMN process to provide meaningful task labels online from high dimensional input.

**Domain Description.** (*Mysterious Bag of Coins*) Our first domain is a sequence prediction game involving a predictor, an opponent and a bag of $m$ biased coins. Flipping the $i$th coin involves sampling a value from a parametrized Bernoulli distribution $\mathcal{B}(\theta_i)$, with $\theta_i \in [0, 1]$ for $1 \leq i \leq m$. The predictor knows neither how many coins are in the bag, nor the value of the $\theta_i$ parameters. The data is generated by having the opponent flip a single coin (the choice of which is hidden from the predictor) drawn uniformly from the bag for $X \sim \mathcal{G}(0.005)$ flips, and repeating, where $\mathcal{G}(\theta)$ denotes the geometric distribution with success probability $\theta$. At each time step $t$, the predictor outputs a distribution $\rho_t : \{0, 1\} \to [0, 1]$, and suffers an instantaneous loss of $\ell_t(x_t) := -\log \rho_t(x_t)$. Here we test whether the FMN process can robustly identify change points, and exploit the knowledge that some segments of data appear to be similarly distributed.

(*A Fistful of Digits*) The second test domain uses a similar setup to The Mysterious Bag of Coins, except that now each observation is a 28x28 binary image taken from the MNIST [15] data set. We partitioned the MNIST data into $m = 10$ classes, one for each distinct digit, which we used to derive ten digit-specific empirical distributions. After picking a digit class, a random number $Y = 200 + X \sim \mathcal{G}(0.01)$ of examples are sampled (with replacement) from the associated empirical distribution, before repeating the digit selection and generation process. Similar to before, the predictor is required to output a distribution $\rho_t : \{0, 1\}^{28 \times 28} \to [0, 1]$ over the possible outcomes, suffering an instantaneous loss of $\ell_t(x_t) := -\log \rho_t(x_t)$ at each time step.

(*Continual Atari 2600 Task Identification*) Our third domain consists of a sequence of sampled Atari 2600 frames. Each frame has been downsampled to a $28 \times 28$ resolution and a 3 bit color space for reasons of computational efficiency. The sequence of frames is generated by first picking a game uniformly at random from a set of 45 Atari games (for which a game-specific DQN [16] policy is available), and then generating a random number $Y = 200 + X$ of frames, where $X \sim \mathcal{G}(0.005)$. Each action is chosen by the relevant game specific DQN controller, which uses an epsilon-greedy policy. Once $Y$ frames have been generated, the process is then repeated.

| Algorithm | Average Cumulative Regret | Algorithm | Average Per Digit Loss |
|:---:|:---:|:---:|:---:|
| KT | $783.86 \pm 7.79$ | MADE | $94.08 \pm 0.05$ |
| PTW + KT | $157.19 \pm 0.77$ | PTW + MADE | $94.08 \pm 0.05$ |
| FMN + KT | $148.43 \pm 0.75$ | FMN + MADE | $\mathbf{86.12 \pm 0.28}$ |
| FMN$^*$ + KT | $\mathbf{147.75 \pm 0.74}$ | Oracle | $82.81 \pm 0.06$ |

Figure 4: (Left) Results on the Mysterious Bag of Coins; (Right) Results on a Fistful of Digits.

**Results.** We now describe our experimental setup and results. The following base models were chosen for each test domain: for the Mysterious Bag of Coins (MBOC), we used the KT-estimator [13], a beta-binomial model; for A Fistful of Digits (FOD), we used MADE [9], a recently introduced, general purpose neural density estimator, with 500 hidden units, trained online using ADAGRAD [8] with a learning rate of 0.1; MADE was also the base model for the Continual Atari task, but here a smaller network consisting of 50 neurons was used for reasons of computational efficiency.

*(Sequence Prediction)* For each domain, we compared the performance of the base model, the base model combined with PTW and the base model combined with the FMN process. We also report the performance relative to a domain specific oracle: for the MBOC domain, the oracle is the true data generating source, which has the (unfair) advantage of knowing the location of all potential change-points and task-specific data generating distributions; for the FOD domain, we trained a class conditional MADE model for each digit offline, and applied the relevant task-specific model to each segment. Regret is reported for MBOC since we know the true data generating source, whereas loss is reported for FOD. All results are reported in nats. The sequence length and number of repeated runs for MBOC and FOD was 5k/10k and $2^{21}$/64 respectively. For the MBOC experiment we set $m = 7$ and generated each $\theta_i$ uniformly at random. Our sequence prediction results for each domain are summarized in Figure 4, with 95% confidence intervals provided. Here FMN$^*$ denotes the Forget-me-not algorithm *without* the complexity reducing techniques previously described (these results are only feasible to produce on MBOC). For the FMN results, the MBOC hyper-parameters are $k = 15, \alpha = 0, \beta = 0, c = 4$ and sub-sample sizes of 100; the FOD hyper-parameters are $k = 30, \alpha = 0.2, \beta = 0.06, c = 4$ with sub-sample sizes of 10. Here we see a clear advantage to using the FMN process compared with PTW, and that no significant performance is lost by using the low complexity version of the algorithm.

Digging a bit deeper, it is interesting to note the inability of PTW to improve upon the performance of the base model on FOD. This is in contrast to the FMN process, whose ability to remember previous model states allows it to, over time, develop specialized models across digit specific data from multiple segments, even in the case where the base model can be relatively slow to adapt online. The reverse effect occurs in MBOC, where both FMN and PTW provide a large improvement over the performance of the base model. The advantage of being able to remember is much smaller here due to the speed at which the KT base model can learn, although not insignificant. It is also worth noting that a performance improvement is obtained even though each individual observation is by itself not informative; the FMN process is exploiting the statistical similarity of the outcomes across time.

*(Online Task Identification)* A video demonstrating real-time segmentation of Atari frames can be found at: `http://tinyurl.com/FMNVideo`. Here we see that the (low complexity) FMN quickly learns 45 game specific models, and performs an excellent job of routing experience to the appropriate model. These results provide evidence that this technique can scale to long, high dimensional input sequences using state of the art density models.

## 5 Conclusion

We introduced the Forget-me-not Process, an efficient, non-parametric meta-algorithm for online probabilistic sequence prediction and task-segmentation for piecewise stationary, repeating sources. We provided regret guarantees with respect to piecewise stationary data sources under the logarithmic loss, and validated the method empirically across a range of sequence prediction and task identification problems. For future work, it would be interesting to see whether a single Multiple Model-based Reinforcement Learning [7] agent could be constructed using the Forget-me-not process for task identification. Alternatively, the FMN process could be used to augment the conditional state density models used for value estimation in [18]. Such systems would have the potential to be able to learn to *simultaneously* play many different Atari games from a single stream of experience, as opposed to previous efforts [16, 18] where game specific controllers were learnt independently.

## Footnotes

† indicates joint first authorship.

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
