[Supplementary Material]

# Appendix: The Forget-me-not Process

## 1   Source code

A reference implementation of both the *Partition Tree Weighting* and *Forget-me-not* algorithms as applied to the *Mysterious Bag of Coins* domain is provided at:

<div align="center">

`https://github.com/jwveness/forget-me-not.`

</div>

## 2   Additional Experimental Details

### 2.1   Hyper-parameter Selection for the Fistful of Digits results

We performed a hyper-parameter sweep over the $\alpha$, $\beta$ and $c$ parameters, with the reported values selected so as to minimize the average per digit loss. The size of the model pool $k$ was chosen to be as large as possible given system memory constraints.

### 2.2   Optimization details for the Fistful of Digits results

The AdaGrad state is maintained separately per MADE model. Cloning a model copies both the weights and the AdaGrad state. A gradient step is taken for each model active in a mixture for each sample seen. These individual gradient steps can be combined into a single mini-batch operation when using the complexity reduction technique described on the final line of the *Complexity Reducing Operations* subsection.

## 3   Additional Details on Theoretical Results

### 3.1   Interpretation of Proposition 1

The constraints on the $g$ function should be interpreted as the base model "having known regret properties"; many methods satisfy this, for example those that have their cumulative regret bounded by $O(\log n)$ or $O(\sqrt{n})$. The restriction to memory bounded sources is a technicality to do with using finite length segments; for example, although there are regret bounds for algorithms that model infinite $k$-Markov sources, our result would not hold (though it would for finite $k$). In practice the majority of base models one would reasonably consider are memory bounded for reasons of computational efficiency.