[Reviews · NeurIPS 2016]

Reviewer 1

Summary

The paper extends the Partition Tree Weighting "meta-algorithm" by Veness (et al.) by adding a memory that allows to reuse existing sources (rather than forgetting them). The result is called the "Forget-me-not" (FMN) process, which is presented in detail in the paper, along with some theoretical properties, a discussion, and experimental results.

Qualitative Assessment

This is an excellent paper that is very well written, makes a clear and interesting contribution and links well to existing literature. The presentation is concise and precise, though in places perhaps a tad more formal/mathematical than necessary. I recommend accepting this paper.

Confidence in this Review

2-Confident (read it all; understood it all reasonably well)


Reviewer 2

Summary

The paper is concerned with modelling piece-wise stationary time series processes, with the sequence prediction problem in mind. It is assumed that a model is available that works for stationary processes, and this model has to be adapted for use in the piece-wise stationary setting. The paper builds on the previous work [14] that has proposed a CTW-like tree approach. The proposed extension concerns the setting where it is further assumed that on some of the segments the distribution of the process may be the same. Some theoretical considerations are presented, and empirical evaluation in three different simulated domains is given.

Qualitative Assessment

It is not clear what exactly is the improvement compared to [14]. I understand in what direction an improvement was sought, but I do not see what is achieved. Specifically, as the authors state (lines 206-209), there is no theoretical improvement in the setting targeted. Since the experimental results were only carried out in simulated settings, and since the problem studied is a well-known one with many known applications, I do not consider the experimental results of sufficient interest to warrant a publication (regardless of the values in the tables). Besides, if I understand correctly, the authors do not position the experimental results as the main contribution either. Lesser comments: I think it could have helped the paper if the theoretical results of [14] were stated more clearly, and the assumptions made for those results to hold were also explained and justified. Specifically, from the paper it is not clear while any assumptions on the given stationary model rho are necessary at all, while in Proposition 1 suddenly some strong assumptions appear - apprently, these are carried over from [14]. The notation is also confusing at parts. For example, the notation \rho[..] (square brackets) is not introduced. As a minor comment, the title of the paper is confusing, as the term "The Forget-me-not Process" refers in fact to an algorithm, not to a process.

Confidence in this Review

3-Expert (read the paper in detail, know the area, quite certain of my opinion)


Reviewer 3

Summary

The authors study online probabilistic sequence prediction for piecewise stationary sources. Their focus is on situations where the sources may often repeat. They introduce the Forget-me-not (FMN) process, which is a nonparametric meta-algorithm for this task. This basically builds on the Partition Tree Weighting (PTW) meta-algorithm and incorporates a memory of previous model states, which is used to improve performance on repeated tasks. More precisely, they consider a two-level hierarchical process, with the first level being PTW, and the second level performing model averaging over a growing set of stored base model states. Theoretically, the authors provide a regret bound for FMN which is of the same asymptotic order as PTW. Empirically, the authors evaluate their algorithm across three test domains, and show that FMN performs well, better than PTW. This gain is more prominent in some settings than in others, but in any case the experiments show that running the FMN meta-algorithm is indeed worthwhile for repeating sources as it achieves lower regret than PTW.

Qualitative Assessment

The FMN meta-algorithm that the authors introduce, analyze, and evaluate experimentally is a solid contribution and will be of interest to the NIPS community. It builds on the previous PTW meta-algorithm in a logical way to incorporate repeating sources. The ideas are simple but well executed; the improvements are reasonable and worth doing if one suspects that the sources are indeed repeating.

Confidence in this Review

2-Confident (read it all; understood it all reasonably well)


Reviewer 4

Summary

This paper studies predicting in the time domain where the probabilistic source switches but a single source may be used for multiple intervals. The authors define the problem, provide algorithm, and analysis.

Qualitative Assessment

My main reservation about this work is the lack of motivation. It is not clear what is a real world scenario that this work is working towards solving. Therefore, it is hard to justify the definitions and the model being developed. • The abstract is missing a motivating example. For example, in lines 33-34: it is claimed that there are many scenarios in which the same regime repeats multiple times however not even a single example is presented. • Line 60: the notation is non-standard, in this paper the interval (a,b) is the closed interval. I am more used to (a,b) marking the open interval and [a,b] marking the closed interval • I think the notation can be simplified by saying that the partition of 1..n is a set a_1,...,a_k such that a_1=1, a_k=n and the a_i's are ascending. In this notation you can use the half open intervals starting at [a_i, a_{i+1}). • The definition of the generating model (between lines 69 and 70) does not have any constraint (or prior) on the partition. Is this by intent? • Line 73: the definition of the repeating source requires that there are two segments that use the same source. This is a very weak constraint since if I have n points (where n is large) it is sufficient that 2 points use the same source to qualify. • Lines 145-146: what does it mean “always better”? Even if the data is generated by a certain source, there is a certain (small) probability that the data seen so far is non-typical. • Line 163: “given an an arbitrary string” – delete one of the an’s. • Line 113: according to this approximation, it is possible that most intervals are spurious (when log(n) is large), in what sense is it a good approximation? After reading the responses of the authors I have to, unfortunately, lower my ratings for this work. Even in the response the authors did not supply a concrete motivating example for the setting they define. I find this to be a fatal flow since it is possible to define so many different models but without the guiding light of an application this effort could be useless. I think this work has many merits and I encourage the authors to re-write this paper and submit it again.

Confidence in this Review

1-Less confident (might not have understood significant parts)


Reviewer 5

Summary

The paper introduce a new meta-algorithm called Forget Me Not. This method is for treating piecewise stationary sources having in hand an algorithm with nice regret properties for the stationary case. The meta-procedure has the same warranty than the previous PTW: a logarithmic regret O(log n) for a complexity O(n log n). The novelty of the approach is to incorporate a memory of previous model. In practice, it over performed PTW when the piecewise stationary sources repeat through time.

Qualitative Assessment

Major comments: The novelty of the paper is genuine mostly thanks to the setting of repeated source that provides very nice applications. The theoretical results are disappointing as they do not hold in the context of repeated stationary sources where FMN would outperform any existing algorithm as it is tailored for that. Minor comments: The preliminaries are copied from [14] but are extremely clear and well-written. Eq. (4) is crucial in the analysis to save complexity and in cost. The justification is treated a bit loosely and some clear conditions on \rho would be helpful followed by a discussion on how to check it on models. I do not understand Figure 3: is the horizontal axis the time or different symbols? It seems to me that the actual Figure is just another representation of the full tree modeling the memory. Finally, I do not understand line 8 of Algorithm 1. It seems that then all the b and r are equal to 0 at any time in the algorithm. Typos: p.2 l.60: the notation d should be used only for the depth p.2 l.72: the terminated point y is not defined p.6 Algorithm 1: the index i should not be used twice, l.4 and l.11-12 After the rebuttal: The authors did not suggest any potential modifications of the main theoretical result. I was hoping some new theoretical arguments to obtain a result different than PTW by restricting the result to repeating sources. On the contrary, the authors insist in developing further the application part that was already satisfactory. For the lack of novelty of the theoretical result and no suggestion of any improvement, I rate the theoretical level of the paper as low.

Confidence in this Review

2-Confident (read it all; understood it all reasonably well)


Reviewer 6

Summary

The authors propose a scheme suitable for the predictions of piecewise "stationary" sequences, where here "stationarity" merely means that the sub-sequence corresponding to each "stationary" component is efficiently learnable by a given prediction scheme. The main challenge in this setting is therefore efficient identification of the boundaries between these components. The paper builds upon Bayesian mixing schemes and particularly PTW, and its novelty is in being able to exploit repetition within the components. The authors present an algorithm, along with several practically crucial complexity reduction heuristics, as well as several experiments studying the efficacy of their approach.

Qualitative Assessment

Overall, I believe this is a good paper - the algorithm appears non-trivial and innovative, and the experiments are convincing. However, the paper also has, in my opinion, several shortcomings: 1) There were several technical aspects of the algorithm and its implementation that I could not understand: A) In general, have found the pseudocode for the algorithm hard to follow, despite reading the entire paper; I would recommend more verbal description (perhaps as comments next to the pseudocode) and if possible, less notation. B) The combination of the method with MADE is not clear to me. In particular, when and for which models gradient steps are taken? Are mini-batches used? Does the AdaGrad optimizer keep a separate state for each model? While I understand that the details of the training of the prediction models can be abstracted away while considering the mixing scheme, I think presenting these details when describing the algorithm is important and can make the presentation far more accessible. C) Two aspects of the complexity reducing heuristics were not clear to me: I) If rho* is selected as the best performing model for the segment, it seems unlikely for the condition of option 2 of the pruning heuristic to ever happen. In particular, if no sub-sampling is used, shouldn't rho*(s) be greater than xi(s) by construction? II) In option 3 of the pruning heuristic, how exactly is the "age" of a model determined? D) How were the hyper-parameters determined in each experiment? 2) The efficacy of the approach in exploiting repetition is not established theoretically. Instead, the guarantees of PTW are shown to be inherited, and efficacy for repetition is deferred to experimental studies. Moreover, the theory doesn't seem to consider the complexity reducing heuristics, without which the approach has very limited practical value. I believe that the "Theoretical Properties" paragraph of the paper is sufficiently weak to be relegated to supplementary materials, and the freed up space can be used for more explanations of other aspects of the paper. 3) Motivation for the work is lacking in the sense that no convincing applications are described. The only "real-world" application for the proposed algorithm mentioned in the paper is multiple-task reinforcement learning, with a stream of different Atari games given as an example. However, it is not clear to me why should the identity of the game played be hidden from the agent learning multiple Atari games. Note that while I feel motivation is lacking in the presentation, I do believe the suitable applications exist, and answered question 7 of the review accordingly. I believe the authors should provide additional motivating examples in the introduction to the paper. Here are some additional comments and suggestions: a) The abstract refers to the innovative aspect of this work with only one word ("repeating"). I believe this should be expanded to at least one sentence (I only understood that this is the innovative aspect after reading the introduction). b) In the denominator just below line 86, I believe it should be rho instead of nu c) In line 129, I believe it should be rho[x_{a_{i-1}:b_{i-1}}] rather than rho[x_{a_{i}:b_{i}}] d) In line 166, is "marginal probability" the right term here? Shouldn't it be "joint probability"? e) In line 177, the assumption that prediction takes O(1) time is perhaps not detailed enough when considering neural networks as statistical models; instead, it should be interesting to count exactly the number of forward and backward passes, as well as parameter updates, that the algorithm uses. f) The use of a generative model such as MADE in an online fashion is, to my knowledge, quite non-standard. It should be interesting to see a graph of the commutative regret as a function of time in order to get more insight into the learning dynamics and convergence behavior of the model. g) In line 259, the oracle is reported for the FOD domain, not the MBOC.

Confidence in this Review

2-Confident (read it all; understood it all reasonably well)